# Reflections on Service Learning for a Circular Economy Project in a Guatemalan Neighborhood, Central America

**Peter A. Kumble**

Faculty of Environmental Sciences, Department of Land Use and Improvement, Faculty of Environmental Sciences, Czech University of Life Sciences, Prague 16500, Czech Republic; kumblep@fzp.czu.cz

**Abstract:** The research presented in this paper explored multiple objectives. First, what are the requirements for establishing a new composting business that embraces the principles of circular economy? Second, how can employment opportunities for at-risk youths from the most impoverished neighborhood in Guatemala City be created, while adhering to the tenets of social sustainability, of which human rights is the corner stone? Third, what were the requirements involved in making compost in the challenging climatic conditions of Guatemala City? And finally, from an educational perspective, how can this be taught to university students incorporating community service learning in its pedagogy, coupled with the model of action research? What are the obstacles to overcome when initiating a startup business, balancing what appeared to be a mix of complex economic, environmental, and social elements? These three elements are the recognized pillars of sustainability, and as such, there existed a great opportunity to meld the principles of circular economy, community service learning, and action research within the context of putting theories into practice. This applied research attempted to explore how effectively this could be accomplished in Guatemala while overcoming complex cultural, environmental, and economic barriers.

**Keywords:** circular economy; Guatemala; action research; social sustainability; community service learning

---

## 1. Introduction

Located in Central America, Guatemala is bordered by the Pacific Ocean to the west and the Caribbean Sea to the east and shares international borders with Belize to the northeast. Geographically, the country is unique, as it touches both the Pacific Ocean and the Caribbean Sea. Neighboring countries include Honduras and El Salvador to the southeast and Mexico to the north and northwest (Figure A1).

The capital, Guatemala City, is situated centrally in the southern portion of the country at an altitude of 1500 m (nearly 5000 feet), resulting in cool evening temperatures during the summer months. According to the United Nations Department of Economic and Social Affairs (DESA) Population Division, Guatemala City has approximately 1,202,536 residents; this number triples during the week when low-income people from rural areas travel by bus into the city for work [1]. However, many low-income people also live within the city and are concentrated into 22 specific neighborhood and employment zones. The social and economic classes of those who live and work within each of the zones typically distinguish the demographic and economic character of each zone.

Zone 3 is the most impoverished of all the 22 zones where dwellings have been built upon early vestiges of the nearby *basurero*—landfill (Figure A2). People who work in the *basurero*—not for a living wage but rather to pick through the unsorted trash to support their fledgling substance have erected a

neighborhood using concrete blocks and plywood, placed upon a layer of soil and trash to construct shanty houses [2]. It is here that families are caught in a cycle of poverty as they manually sort through the trash as it is dumped from trucks. Many are in search of either plastic and/or glass that may be redeemed at recycling facilities in the city, discarded food to eat, or discarded household items to claim as their own; the dangers of this existence are significant [3].

The *basurero* is located within a 16.2 hectare (40 acre) ravine that receives an estimated one-third of Guatemala's waste each day. Although municipal authorities claim that the site has reached its capacity and must be shut down [1], the waste stream continues to flow in, primarily because an alternative site has not been identified. During the five-to-six month rainy season (May through October), the unstable land frequently collapses [4], engulfing some of those who are sorting through the trash (Figure A3). Often, the bodies are never recovered because of the natural composting process of the organic waste contained in the decomposing trash. The organic waste produces high levels of methane gas, which caught fire in 2005 resulting in contaminated air and heavy smoke and claiming numerous lives over a period of many weeks. The *basurero* caught fire again in January 2014; fire fighters had great difficulty in extinguishing the blaze due to the aforementioned flammable methane gas and hazardous terrain. In addition, the greenhouse gas emitted from the waste typically causes cancer and tumors among those working in the *basurero*; these laborers have little or no access to basic health care.

The research conducted in this paper attempted to determine if one could make compost for the use in the parks and planting beds managed by Guatemala City from the green waste generated by a large central market (CENMA), thus diverting it from disposal in the landfill. In addition to selling the finished product locally, the research attempted to determine if the finished compost might have a regional market by using the trucks that brought fruits and vegetables to the market and sending it back to those very farms to augment their soil. In theory, this goal of developing a closed-loop system supported the principles of circular economy (CE), a term that first appeared in the literature as part of a study by Pearce and Turner (1990) [5]. Their research, as later reported by Anderson (2007), attempted to make a connection between a variety of complex production activities at both an economic and environmental scale [6]. This is an approach broadly acknowledged as an appropriate strategy for reducing waste and improving manufacturing efficiency in urban contexts [7–9], often referred to as a "closed-loop system", first discussed by Boulding (1966) and later by Leontief (1991), whereby one repeatedly uses the same resource within a closed-loop [10,11]. This principal assumes that a finished product or the byproduct of some sort of manufacturing can then return to the source from where the raw materials originated, with the intention to then improve growth of that material for greater future use. Achieving this goal is perhaps most recognized in different forms of agricultural production where green and animal waste is directly recycled into the soil as fertilizer. This type of closed loop system can be more challenging within the manufacturing industries; however, this is very much a case-specific comparative analysis. The United Nations Environment Program (2012) stated that within a circular economic system, nutrients and associated resources circulate and remain within different "biospheres and product systems" [12]. However, when measuring ecological efficiency within the context of CE, the diverse and complex nature of what we make and how we distribute and consume it can cloud what one might think is a clear connection between the two—ecological efficiency and CE. More affluent populations demand more resources to meet their social and economic needs [13–15], which tend to threaten depletion of finite resources. Others found that circular economy is a concept that has achieved recognition by academics, governments, and the private sector [16]. Merli et al. (2018) explored how scholars might better understand the complexities of CE through their extensive literature review of 565 articles. The literature found that while CE works to overcome what they refer to as the "take–make–disposal linear pattern of production and consumption", the real goal is to maintain materials, resources, and their associated products in the economy as long as possible [17]. Their review confirmed that a very close relationship exists between CE and the concepts of sustainability; however, they also found that scholars seldom consider social and institutional implications of the implementation of CE at the environmental and economic level. Lyle (1985) worked

to overcome this disconnection through his applied research in the 1980s, presenting the concept of regenerative design in his book, *Design for Human Ecosystems* [18], which effectively illustrated closed-loop systems for achieving waste recycling, energy production, and a more efficient use of resources. This continues to be demonstrated in application at the Center for Regenerative Studies at Cal Poly Pomona, California.

Yet, CE remains a disputed or contested concept [19,20], perhaps due to many different definitions of what it means, likely due to its interdisciplinary theory and application. Kirchherr et al. (2017) analyzed 114 definitions of the term and developed a meta-definition: "a CE describes an economic system that is based on business models which replace the end of life concept with reducing, and alternatively reusing and recycling materials in production, distribution, and consumption processes ... " [21]. Perhaps the challenge not only lies within its definition but in how to apply business and policy models to a myriad of legal, technical, and environmental situations. Geissdoerfer et al. (2017) found that CE is now seen as a possible solution for achieving the principles of sustainable development [7]. They found that the sustainable "performance" of circular business models and circular supply chains are necessary to implement sustainable development based on case studies and a review of the literature. De Jesus and Mendonca (2018) reported that CE should be viewed as a "motivational and inspirational compass" and proposed the concept of eco-innovation (EI) as a way to better achieve positive impacts on the environment and society. They found that EI has an appropriate role in the move toward CE on a broader scale, in that EI proposes new or improved socio-technical solutions to preserve resources, reduce environmental degradation, and improve the recovery of waste [22]. Others have proposed the need to have measurable indicators of the longevity of a resource as a measure of CE efficiency. Yet, what should those indicators be and how do they vary, given the enormous range of product(s), resources, and other measures of CE? Figge et al. (2018) argue that longevity and circularity are necessary for sustainable resource use, but thus far, there are no applied measures that combine both approaches [23]. They argue that the number of times that a resource is used and reused must also consider the time and duration of that use (longevity) and proposed a complex combination matrix to measure both. This is unique, as it applies both quantitative and qualitative parameters to the concept of CE. Recent trends in the agro-food industry have argued for fair trade and responsible consumption as a necessary measure of sustainable development within CE [24]. With most fair-trade models, consumer demand for products produced in developing countries are marketed and then consumed at a much broader worldwide scale. This forces one to question if achieving true CE is possible within a global scale of consumerism, at a high level of production. However, in practice, achieving a process of sustainable CE can in fact be realized on multiple scales, be that the production of coffee, clothing, or compost. In recent years, financial markets and their associated intermediaries have advocated investing in "green projects" that contribute toward the development of a more environmentally sustainable economy. Falcone el al. (2018) reported on the trend toward ethical investing and socially responsible projects [25], whereby financial institutions are now making access to capital easier while venture capitalists find that green projects add value to companies. They found that there has also been a trend toward funding radical green innovation. Some may call this little more than green-washing business as usual, while others see it as a shift in how business is now being done because it costs less, requires less energy, and achieves socially measurable outcomes. Thus, linking green finance (GF) to CE where waste management is achieved through superior design of materials, products, and associated systems [26] has been achieved particularly within the biomass production sector. The United Nations Environmental Program (2011) found that the movement toward a low carbon and climate resilient economy required investment in green sectors of business [27]. To place this within the context of green economies, such as biomass production, investment in renewable energy, or other CE related industries, e.g., making compost from green waste in Guatemala, the necessary investment worldwide will equal 2% of the GDP between 2010 and 2050 [28]. This is a significant sum of financial investment, but it is necessary to slow the impacts of climate change on a worldwide scale. Considering this, how does CE fit in within the context of this paper—making compost—and how does it begin to form a

guiding principle for local municipal governments to follow? While some countries have adopted a top-down centralized approach, such as China [28], others emphasize a more localized bottom-up policy direction [29], more typically used in municipal and regional planning. Perhaps the difference is again associated with the scale of CE.

As such, this research had multiple objectives: What are the requirements for establishing a new composting business that embraces the principles of CE? Could this initiate a break from the cycle of poverty, create opportunities for a future livelihood, while adhering to the tenets of social sustainability [30], of which human rights is the corner stone [31]? From a more technical and perhaps scientific perspective, what are the challenges involved in making compost in the climatic conditions of Guatemala City, high in altitude, dry during portions of the year, and inundated by rainstorms during other times? Finally, from an educational perspective, how should one teach this to university students through a new class that incorporated community service learning (CSL) in its pedagogy coupled with the model of action-based research?

Thus, the research described in this paper explored techniques that could provide living-wage jobs for unemployed and disadvantaged young people who subsist in one of the most impoverished neighborhoods of Guatemala City. This was achieved by initiating a startup business, balancing what appeared to be a mix of complex economic, environmental, and social priorities. These three elements are recognized key pillars of sustainability [32], and as such, there existed a great opportunity to meld the issues at hand, while providing an appropriate context for putting theories into practice. The important movement toward sustainable development within society has been part of the broader dialog dating to 1992 and the Earth Summit in Rio de Janeiro and again repeated at the World Summit for Sustainable Development in Johannesburg in 2002 [33]. However, bridging the gap from theory to actualized process and practice brought forth the challenges in truly understanding the operational relationships between each in application. Boyer et al. (2016) identified the challenges in understanding and managing social sustainability due to its many meanings and gaps in interdisciplinary research [30]. In retrospect, social sustainability, along with environmental factors and key components of economy are in fact very much place-based; however, the challenges in understanding and managing the social aspects proved to be one of the greater challenges. Boyer's research team used the analogy of a three-legged stool as well as the triple bottom line, the 3Es, and the 3Ps—prosperity, planet and people—as better approaches for solving complex world problems rather than completely inventing new approaches. Thus, the rationale for this new research effort seemed to be a win–win scenario; compost created from organic waste diverted from the landfill would be used to amend marginal soil, train workers, create jobs, mitigate an ecological and environmental crisis, and serve as an image of hope and renewal for those whose livelihood is dependent on the *basurero*. This idea would adhere to the concepts of CE and, if successful, would reduce many negative environmental impacts and stimulate other business opportunities, thus adhering to the bottom-up model proposed by Ghisellini el al., 2016 [30]. It is not a new idea as the concept of a closed loop material cycle has been practiced since the dawn of the industrialization in manufacturing [34,35]. Our research question attempted to explore how effectively this could be accomplished in Guatemala while overcoming complex cultural, environmental, and economic barriers.

*Context and Local Situation*

As a research scholar, one approaches a new opportunity with the intention of testing a preconceived idea, anticipating a successful outcome; you identify a problem and hope for a compelling result or solution. Perhaps the flaw in this approach is that you may overlook a hidden gem of a different research idea, initially not obvious. Thus, how should one move forward when what is discovered was completely unexpected? A chance encounter with a remarkable local activist, a bright and highly motivated graduate student, and an ominous neighborhood trapped within a cycle of poverty—these factors all contributed to an opportunity to produce real and meaningful change to the lives of marginalized children living in a squat neighborhood. The importance of remaining

open to all possibilities and paying attention to amazing new connections for collaboration wherever they present themselves led to significant learning opportunities. The results of this effort were used to develop a startup business, using the principles of circular economy and social sustainability to improve the living conditions through job-creation at one of the most impoverished neighborhoods in Guatemala City.

This endeavor began with the Director for Public Works, Department of Construction, in Guatemala City. The original intention involved conducting research that would explore the landscape of urban parks and recreation zones. Because the author was a professor of landscape architecture, the director provided tours of the City's municipal nursery where much of the ornamental plant stock is grown for beautification of the parks and public spaces in Guatemala City. The nursery was also responsible for making compost; however, nearly $300,000 USD was spent each year to produce compost for soil amendment to the rocky and volcanic planting beds prevalent in Guatemala City. A visit was also paid to the city-owned CENMA market; this extensive outdoor market is where much of the fruit and vegetables produced in the country are prepared for sale and distribution both locally and internationally. The CENMA site is enormous (Figure A4), and it essentially functions as a city within a city, employing a broad spectrum of workers and containing numerous small shops where the public can purchase prepared food and other daily necessities.

The author also visited the *basuerto*, the largest municipal landfill in Central America also located within the City (Figure A5). Follow-up meetings with the Director for Public Works confirmed that much of the green waste from CENMA added to the overflowing landfill. With this realization, a proposal was offered as a solution to the disposal of green waste and a reduction in the money that the city spent each year to produce soil augmentation for their parks and green spaces.

## 2. Materials and Methods

Why focus on helping the youth? Minica and France (2008) merged the social components of sustainable development into four key objectives: promotion of education and training; protecting and promoting human health; the fight against poverty; and equity, i.e., not marginalizing one demographic sector [36]. It has been shown that self-empowerment can be achieved through education. According to Knoth (2009), only 69.1 % of Guatemala's population can read, making it the most illiterate nation in Central America. Similarly, more than 80% of the Guatemalan population will never graduate from high school, not as a result of laziness but rather out of the need for contributing to the family income. At present, the high school dropout rate of the impoverished youths from Zone 3 is incomprehensibly high because of the dire need for these children to earn money to support their families. Sustainable part-time jobs are inaccessible for this demographic group from Zone 3 due to their poor educational background. This trend results in elementary and high school students who tend to leave their short-lived scholastic careers to inherit a place in the cycle of poverty [37]. This sobering realization lead us to believe that the social or human component of sustainable development plays perhaps the dominant role, if not the most important, because of the dire need for equity in the evolution of society today.

CENMA, Guatemala City's largest fruit and vegetable market, sends 114.7 cubic meters (150 cubic yards) of organic waste to the *basurero* daily.

Although other entrepreneurs have tried to capitalize on reducing the waste from CENMA, Guatemala City chose to work with graduate students from the University of Massachusetts, Department of Landscape Architecture and Regional Planning, under the direction of the author, because of the project team's commitment to social justice and social sustainability. The City granted 0.48 hectares (1.2 acres) of land and a plethora of organic material from CENMA for developing a sustainable start-up composting operation. Our goal was to do this embracing the principles of CE and sustainability in action.

The compost produced would have a guaranteed client. As described earlier, Guatemala City was spending annually nearly $300,000 USD on low-grade soil amendment. As such, a contract was developed concerning the municipal purchase of the compost to amend public-sector landscaping projects. This compost would be purchased from a new University of Massachusetts student-initiated start-up company. This compost could also be sold to local farmers, private landscaping contractors, and homeowners, adhering to the principles of CE.

## 2.1. Design Response

This paper does not intend to discuss the finer details of production of compost—the correct mixing of brown and green waste. A plethora of detailed research exists, and how-to manuals are available that document the art of making compost. However, to fully appreciate the work attempted in Guatemala, it is important to provide a brief overview of how one makes compost. The first step in managing the compost process is to determine what is introduced or added to a compost pile. Note that the use of animal manure or carcasses makes composting much more complicated, as human contact with animal waste can spread diseases. Thus, the work reported in this paper only concerns itself only with the composting of organic plant material. Achieving the correct combination of plant material is important for speeding up the composting process and producing a quality product. The two categories of plant materials are referred to as brown and green material [38]. Brown material, such as wood chips or dry grass stalks, do not break down as rapidly, thus giving compost its light fluffy texture. Green material is fresh, wet, and usually green in color. Green material, such as vegetables or plant biomass, will decompose quickly and is balanced, so to speak, by the brown material which is more stable [39].

The word 'organic' refers to alive, or once living, organisms. It is the flesh, bones, tissues, stems, and bark of plants and animals; organic material is natural and not created by man. For example, manure is a natural fertilizer and can be called organic. Cardboard, although processed by man, comes from trees and is organic unless it contains dyes or is coated with plastic on its surface. Pesticides are usually man-made, cannot be found in nature, and are therefore not organic. Yet, sometimes pesticides are organic because some plants make chemicals naturally in their leaves to protect against insects.

Decomposition is the natural process of organic material breaking down to a more chemically stable state. This process constantly occurs on forest floors in the leaf litter or dead wood from trees and shrubs. Decomposition creates nutrient-enriched humus (a fine organic substance seen in high-grade soil), returns nutrients to the soil, and allows new plants to grow [40]. Decomposed material appears in the top layers of soil as dark in color. Non-organic materials do not decompose or may require many hundreds of years to do so [41].

Commercial composting is a process by which the decomposition of organic material is controlled such that it occurs faster and produces a consistent, quality product. A 'compost pile' refers to a pile of organic material that is decomposing [42]. The stable finished product is called 'compost', which is typically mixed with existing soil, making that soil healthier and more capable of growing plants. As farmers plow soil and cut vegetables from fields, that soil becomes degraded, making it less fertile or less capable of growing plants. Mixing compost into soil replenishes the nutrients, which in turn contributes to increased soil fertility [43].

## 2.2. How It Works

Decomposition of organic matter in a healthy forest occurs through the digestive processes of microorganisms [41]. These microorganisms feed on dead or dying plant material and animals, recycling them back into the humus layer on the forest floor. As these organisms eat, grow, reproduce, and die themselves, organic material is broken down (decomposed) into compost. These tiny creatures are contained in decomposing organic material and do not need to be added to a commercial composting pile.

Microorganisms require food, water, and air to live in the same way humans do. Their food is the organic material in the compost pile. Although some animals, such as worms and snails are beneficial to a compost pile, they do less work to promote decomposition than the microorganisms do.

By providing air, water, and the appropriate mix of organic material, a compost pile can reach its finished state in a predictable amount of time. For example, the naturally occurring composting process for organic material in a compost bin, windrow, or pile, can take approximately 90 to 120 days to occur [44], assuming that the composting material receives the necessary blend of oxygen and moisture and that it is turned periodically [45]. The duration for achieving a finished product can in fact be accelerated significantly by increasing the amount of oxygen that enters the composting material.

### 2.3. Site Design

Initially, a large flat space located immediately adjacent to the CENMA market was promised to the UMass team for designing the compost operation (Figure A6). This was an ideal location, given the proximity to the CENMA market, easy access to green waste, space for loading completed compost onto trucks for delivery to the City and export to regional farms, and most importantly, ample space for the production of compost. Guatemala City provided an aerial photograph and site topography details, and UMass students began to prepare a site plan for the development of the compost plant. However, when the team arrived in Guatemala City and met with the Director of CENMA, we learned that he would not allow this flat space adjacent to the market to be used out of fear that the compost would smell and hinder business within the market area. The Director was only willing to make a much smaller and more challenging site available, located next to the larger property but situated outside of the market-proper (Figure A7). This alternative site is a 0.48 hectare (1.2 acre) parcel of land that borders the eastern edge of the CENMA market property and is separated by a 3.7 m (12 foot) cement wall. The parcel is divided into three sections by smaller 2.4 m (8 foot) pieces of wall. The parcel runs next to a very steep and hazardous embankment that leads to a small stream meandering through the adjacent property. The main access to the site is via a 6.7 m (22 foot) entrance gate. Needless to say, the smaller site was less than ideal and contained many physical obstacles to make it fully useable, such as the very narrow profile and hazardous embankment. It also forced a redesign of the methodology for how compost would be produced.

### 2.4. Compost Cells or Windrows?

There are two different ways of organizing compost piles. Windrows [46] are long piles of compost (Figure A8). They are at least 1.5 m (5 feet) high, with equal width, and are difficult to manage with only manual labor.

Most commercial operations rely on windrows because they are effective when dealing with large quantities of material; however, they require many hectares of land and expensive machinery, such as a windrow turner and tractor to turn the mix the piles, allowing necessary oxygen to enter and accelerate the decomposing process.

An alternative to windrows, the in-cell technique, uses modular structures that hold the compost in place. These systems can be managed with manual labor and do not require machinery. There are several reasons why in-cell composting on the CENMA site was more appropriate. These include the following:

- Manual labor would employ more youths who would shovel the compost from one cell to another to accelerate the decomposition process;
- Cells eliminate the cost of expensive equipment typical of the wind-row system; and
- Cells will keep heavy equipment such as tractors or front-end loaders safely away from hillside edges.

To some degree, the in-cell technique more closely resembles the compost "bin" typically used by homeowners for decomposing kitchen and yard organic waste, (Figure A9). While most people who do home composting have one compost bin/cell, three cells should be employed to be more effective

(Figure A10). A compost mix is started in cell #1: once contents begin to increase in temperature and shrink in size as the green waste decomposes, all of the contents should then be shoveled into cell #2 and a new batch started in the now available cell #1. The act of moving the mix from cell #1 to cell #2 adds oxygen and mixes the contents, similar to what a windrow turning machine might accomplish. Later, the contents from #2 are moved into #3 for completion, and a new batch is begun in cell #1, which has seen its material moved into cell #2, similar to hopping checkers over each other.

There are many ways to construct cells for producing compost. Wooden shipping pallets were selected for use at the CENMA site because they are inexpensive to purchase, are readily available locally, and site conditions with hard rocky volcanic ground prohibit the use of metal stakes or poles with welded wire fence to form a cell. Thus, cell construction can be modified according to any site, the type of materials to be composted, and the effects weather may have upon producing compost.

For a newly established compost pile, (in cell #1), the temperature will increase during the initial weeks while the size of the pile decreases. This indicates that the compost process is successfully occurring. Table A1 illustrates cell-monitoring data for one of the two test cells constructed initially to chart the time and temperature, in addition to odor (smelliness). Approximately one month after the pile is made, the temperature will drop, and the pile size will stop decreasing. This is when the pile needs to be turned or shoveled into cell #2. Turning the pile involves taking the material out of the cell and mixing it up; it is important to get material from the middle of the pile to the outside layer.

As the microorganisms feed on the organic material. They consume all that is around them. Because they cannot move very far, it is important to mix the pile and put them in contact with new organic material for consumption. Turning a pile also introduces new oxygen, but this oxygen is quickly used and is not the primary function of turning. Oxygen gets inside a compost pile through proper ventilation and pore space (voids in the mix).

In total, the site physically accommodated 409 cells and produced 1483 cubic meters of compost a year. The cells were 1.8 m (6 feet) deep, 1.5 m (5 feet) wide, and 1.2 m (4 feet) tall. An estimated 140 cubic meters of compost production was stored on-site each month. The benefit of storing the finished compost is two-fold: first is the close proximity to the entrance of the site; second its location along the portion of the wall that has the natural loading dock, enabling trucks to back in, load material, and transport the product to its final destination.

*2.5. Waste Separation*

The general public typically does not fully understand the importance of separating organic waste from non-organic waste. The UMass experience at the CENMA market has shown this to be true as the organic waste from CENMA also contained a tremendous amount of un-compostable trash. The public must be educated on this issue if composting organic waste is to be successful. Since most of the waste collected for this site originates from the CENMA market area, education should begin there. Clearly marked barrels were later placed wherever trash is disposed of. The organic waste barrel should not be easily mistaken for a non-organic waste dump container. Because this was a new concept in Guatemala, simply placing different colored barrels side by side clearly conveyed which are intended for organic material and which should be used for all other waste material—trash. The compost facility received 20 cubic meters of raw materials daily: (1/3 organic waste, 1/3 wood chips, 1/3 cardboard).

## 3. Results

While this project and its intended outcome might appear to be achievable, the capital requirements and structural organization needed to begin such a venture were significant, particularly in a challenging Guatemalan economy. The project began as a collaborative effort between Masters of Landscape Architecture student Travis Shultz and the author, resulting in a new graduate class at the University of Massachusetts Amherst Department of Landscape Architecture and Regional Planning. The class would train students on how to make compost and then start a new non-profit company–in Guatemala–built upon the pillars of sustainability.

This new class incorporated key elements of CSL with action-based research in its pedagogy. Entitled "Applied Field Studies in Guatemala", the class attempted to ask and answer the following question: How can we, as aspiring landscape architects and regional planners, connect the passions of our hearts, the skills we have acquired thus far in our educational careers, and the needs of the world? The focus of the class was on the research, design, and application of a specific project: the start-up of a municipal composting facility in Guatemala City. The class included lectures, presentation of research, field trips to visit commercial compost operations, and educational site design. The goals were multi-fold: experiment with pedagogical education through the lens of CSL, focusing on the Zone 3 neighborhood in Guatemala City; teach students how to write effective grant applications; learn how to make compost; construct test compost cells in Guatemala to be monitored weekly; learn to speak Spanish; and then teach students how to begin a start-up business employing principles of CE at the local municipal level [47]. What we explored are the necessary tools for implementation in Central America; and what does it really mean to start a company based on the recognized three pillars of sustainability [24]? The team discovered that starting a new business at this admittedly small scale did not create obstacles for basing all business models on the principles of sustainability. The endeavor would achieve critical areas of social sustainability through employing at-risk youth who had marginal future prospects in their lives. Diverting green waste from the landfill and making compost with that material for use in municipal planting beds and for sale to local merchants provided a net-positive environmental benefit. Economically, if the City made adequate production space available, we could produce enough compost to meet their needs and employ 12 or more youth.

The author along with the eight graduate students enrolled in the class and a Boston-based brownfield remediation consultant traveled to Guatemala City in the spring of 2009 for 10 days of site design and practical work. Once in Guatemala, the group met with local experts and municipal officials to gather information needed to advance the site design, prepare a business plan, and clarify critical operational information. The team also spent some days working with families in the Zone 3 neighborhood to renovate and reconstruct roofs and other parts of their shanty homes, with the intention of developing a one-to-one connection with the real "client".

As mentioned, the class and its supporting fieldwork were built upon the principles of action-based research. This is an idiom used by educators; yet, it is an approach sometimes surrounded by a lack of clarity in practice, similar to that of CE. Tomal (2003) grouped together the following techniques as a possible definition of action research: field research; a collection of primary research data; and research in which the research question later follows the initial findings [48]. All of these statements combine to form a rational definition; however, a more thorough characterization is needed to clarify why action-based research is one of the most practical, rational, and efficient methods of conducting applied research by educators. In Tomal's book, *Action Research for Educators* (2003), he clarifies that action research is not based upon quantitative methods requiring statistical analysis. It is also not qualitative, requiring enquiry with extensive narrative explanations. "Action research is a systematic process of solving ... problems and making improvements" [48]; the author's experience in Guatemala validated Tomal's theories.

Greenwood and Levin (1998) define action research as "social research carried out by a team encompassing an action researcher and members of an organization or community seeking to improve their situation" [49]. The authors make a powerful statement that "social research generating results void of action is counter-intuitive", and they claim that "action is the only sensible way to generate and test knowledge" [49]. The intent of this process of generating information and then testing it in an applied context aims to bridge theory and practice in such a way as to involve a diversity of groups, each contributing their skills to increase the sustainability of communities and organizations.

Sagor (2000) put forward that action research is "a disciplined process of inquiry conducted by and for those taking the action" [50]. For these reasons, the proposed Guatemalan Field Studies class embraced tenets of action-based research and incorporated many key elements of CSL with the aim of

creating transformative learning experiences for both the university students and youth from Zone 3, with the proposed composting company embracing CE theory.

CSL is an academic model for merging education and social justice [51]. It provides a basis on which students benefit from the real-world applications of their academic training, and with this a community benefits from the pro bono time and skills offered by students. Although there are many organizational and logistic components to deal with in any CSL endeavor, the results almost always outweigh the initial efforts invested in the experience. In a study conducted by Westheimer and Kahne (2004), students who had completed a CSL class "expressed excitement at the prospect of getting involved in ways they did not know were available to them before their experience. The curriculum also developed students' desire to participate in civic affairs and gave them a sense that they could make a difference in the lives of others" [52]. Forsyth et al. (2000) asked *"Is Service Learning Worth It?"* and concluded that CSL "has the potential to provide the kinds of skills and approaches that are essential for the professions, particularly the design professions, if they are to retain their relevance" [53]. If faculty, students, and the community are informed beforehand of the challenges concerning the process, many foreseen and unforeseen issues can be more easily pacified as they arise. An important but often overlooked aspect of CSL is self-reflection throughout the process. Simple journaling, surveying, and discussing one's experiences can be the difference between a good class and a semester that challenges one's beliefs, reforms ideals, and redirects lives.

## 4. Discussion

The UMass students in association with locally based faith organization *Vida Joven* (Young Life) were successful in establishing a new business enterprise called AbonOrgániCo, based in Guatemala City. The mission of the company was to supply necessary part-time jobs to at-risk youth from Zone 3 and the surrounding communities. AbonOrgániCo was incorporated in Guatemala as a not-for-profit corporation; excess capital earned would be invested back into the business. As mentioned above, AbonOrgániCo began under the umbrella of Vida Joven in order to facilitate a quick start-up process for the business using recognized local support for the Zone 3 neighborhood. Under the not-for-profit designation, the business did not incur a tax liability in Guatemala. The director of Vida Joven also served as the interim director of AbonOrgániCo. Under the director were a site manager and a supervisor. The supervisor oversaw the work of the employed at-risk youth. For this type of operation, total quality management was essential to the success of the company. Marketing was not necessary for the product, as it was sold to the largest customer, Guatemala City's municipal government. However, promotional strategies were developed to reach other smaller-volume customers (Table A2), testing the theories of CE.

*Technical and Organizational Challenges*

In the short-term, AbonOrgániCo was located on a tract of land adjacent to CENMA; however, this site was not ideal for two reasons. It was not large enough to produce compost that would meet municipal demand, and it was much too small to utilize the entire supply of organic waste generated daily by CENMA. Workers were responsible for the daily activities necessary to operate the business. This included mixing raw materials, turning compost piles, and taking moisture and temperature measurements of the piles. Ten to twelve workers were initially employed. The workers were selected from a pool of at-risk youth within the city who were in need of part-time employment. Only candidates who were currently enrolled in a secondary school were employed. This is directly in support of the mission of AbonOrgániCo, which is to employ youth on a part-time basis so that they may receive an education and break out of the cycle of poverty. Each of the young workers/employees were taught how to open a bank account locally so as to receive their weekly pay. Sponsors from the USA matched each dollar earned by the local kids with a donation of equal pay, thus doubling the money earned.

Guatemala City's municipal government (MUNI) used 15,000 cubic meters of low-grade soil to landscape the area along their roads and scenic-ways projects. For these purposes, an ideal ratio of compost to soil would be 1:1. This means that the municipality had a demand for roughly 7500 cubic meters annually. Because the current 0.48 hectare site adjacent to the CENMA market is incapable of meeting this demand for compost, a larger work site would be necessary. To produce the 7500 cubic meters of finished compost annually that Guatemala City would purchase, AbonOrgániCo would need to produce 625 cubic meters monthly, or approximately 31 cubic meters each day. It would seem that the necessary site would need to be roughly six times larger than the size of the current CENMA site. This would equate to about 2.6 hectares (6.4 acres). However, if the more efficient windrow method were utilized on a larger site, the actual area needed to meet the municipal demand is estimated to be approximately 1.6 hectares (4.0 acres). Although the City promised that a larger site would be provided just to the southern edge of CENMA on municipal land, this did not happen.

Another key point when considering an ideal site for AbonOrgániCo is the volume of organic waste that CENMA sends to the landfill each day. This amount equals roughly 138 cubic meters. The current site, and even the 2.6 hectares site discussed above, is vastly insufficient to handle this supply of organic material if all the waste were to be converted into compost. In order to effectively divert all of this waste from the landfill, a site of approximately 5.8 hectares is necessary. Ultimately, the limitations of space had a significant negative impact on the volume of compost that was produced. AbonOrgániCo has never been able to meet the real demand for compost by the Municipal Government of Guatemala City; however, with a larger working production site, this could be achieved in the future.

What we learned from the monitoring data of the two test cells (Table A1) was that the high altitude dry climate of Guatemala City caused much of the moisture in the newly mixed compost (cell #1) to dry out prematurely, resulting in a very slow or even stalled rate of decomposition. This required altering the brown to green mix to increase the green organic volume during the initial mixing of the new compost piles in cell #1.

While poverty will never be completely overcome—and improved living conditions for those from Zone 3 achieved—through a commercial composting business, every step in support of this goal can and will make a tangible, and most importantly a sustainable, difference in the future of children who live in Zone 3 with little future prospects. Each individual who becomes involved in a venture, such as AbonOrgániCo, has a unique skill to contribute, whether it is the desire for developing programs to address social justice, expanding the knowledge-base of composting methods, developing fund-raising opportunities, or connecting people who have these skills. Each part or component of the program for recycling green waste from CENMA, putting to work at-risk youths of Zone 3 neighborhoods, and creating a useable product that was economically viable adhered to the three pillars of sustainably (Figure A11).

Sustainability in practice applies here, regardless of whether one is a proponent of social justice, an entrepreneur in search of starting a sustainable company to help the poor, or a public official determining the level of feasibility of a project, or even a potential financial donor. The intention is that this project will encourage others to follow the passions of their heart to where they intersect with the skills one may have learned professionally or in school.

## 5. Conclusions

During the applied field-learning excursion to Central America, students kept daily journals—a key pedagogical technique used in many CSL classes—and held daily group meetings to ensure communal learning. Each day they shared indelible memories that resulted in personal growth and new understandings of their place within the project and perhaps beyond. Although the spoken Spanish language skills of the students often meant that communication with residents from Zone 3 and other potential vendors in Guatemala was kept at a very basic level, they received respect and support in that we were not perceived as elite scholars but rather as individuals who tried to communicate and connect with locals. Other opportunities for reflection were provided though daily group discussion

sessions each evening while in Guatemala. To quote the summary provided in the thesis reflection prepared by Masters of Landscape Architecture student Travis Shultz, "these meetings began with each member of the team quickly explaining his or her most indelible or memorable moment of the day and ended with a student giving an overview of their life experiences and how they have come to where they are today" [54]. Although this may appear to be merely a "feel-good" exercise, the opportunity to unify the group far outweighs the difficulties and awkwardness of organizing these interactions. Encouraging students to recall their most memorable moments of the day enabled a more lasting impact, as spoken words affirm experiences [53].

Opportunities for people to express themselves without interruption rarely presents itself, and many people even struggle with communicating their own life story, as American culture does not typically allow time for interactions such as these. However, from this experience, each student can now say that they truly know each other, not superficially or for the purpose of putting a document together, but because there is now a shared atmosphere of interest and concern for each other as people within a community. This cohesiveness could be seen in group presentations and group work products—from constructing the test compost cells to preparing an achievable business plan and securing significant grant funding to start the company.

After the trip, the class continued with meetings twice a week for the purpose of producing an operational manual (Figure A12) for how to make compost, summarize the site analysis and field assessment, prepare an operational budget and business plan, design the physical components for a compost operation, and finally, learn how to write funding grants in support of the project. Table A3 shows the table of contents for the operational manual. The manual is extensive with a variety of construction techniques prepared in detail (Figure A13) and can be used to guide the development of a composting business in locations other than just Guatemala City. The group also developed strategies for how the compost could be effectively sold to a variety of customers within Guatemala (Table A2) and prepared different promotional strategies to achieve success (Table A4). The goal of loading finished compost onto the empty trucks that had just finished unloading fruit and produce at CENMA, and then to deliver the compost to the farms that grew the fruits and vegetables, adhered to the theories of CE. This was the motivational and inspirational compass as described by de Jesus and Mendonca (2018) [22]. However, this goal was never realized as the farmers were not willing to pay money for the compost and the truckers were not willing to deliver a product that they would not be financially compensated for without some level of government support. In addition, the composting business was not able to produce the volume of compost needed by Guatemala City Municipal Government. It also could not accommodate all of the green waste generated by CENMA—all primarily due to the inadequately small space allotted by CENMA for the compost operation. Regardless, nearly all of the youth hired by the company did learn a trade and make some money to support their family. On reflection, this was a good beta-test or experiment to determine if compost could be made in the environmental conditions of Guatemala City. The team learned about the spatial requirements necessary to produce a commodity viable for sale to the municipal government, local contractors, and for shipping back to farms. From an educational point of view, the effort did adhere to the principles of CE, EI, action research, and CSL. Considering the complexity of melding so many different educational, environmental, social, and economic theories, the effort was successful. As mentioned, the limiting factor was adequate space for the production of enough compost.

During the final weeks of the semester, the group discussed and analyzed their experiences, focusing on how this work relates to their educational and professional careers in landscape architecture and regional planning. The group concluded their work with a University-wide presentation to professors and peers and graduate student Travis Shultz successfully defended his Masters of Landscape Architecture thesis reflecting on this experience.

In keeping with the mission and spirit of a CSL project, the students found that the level of success that they achieved while working within the marginalized Zone 3 neighborhood of Guatemala City was tremendously powerful. Everyone felt that if eight university students could achieve this level of success in three months, the possibilities for what could be achieved through implementing this concept professionally after graduation are limitless.

The students discovered that if they were to apply the broad range of skills taught in a landscape architectural curriculum, coupled with one's ability to support the underprivileged, the opportunities available have no limits or bounds. Helping and being helped by those in need is not a concept foreign to the professions of landscape architecture and regional planning. It is a genre within the profession that should be taught, learned, and then experienced first-hand to be fully appreciated.

**Funding:** This research received no external funding.

**Acknowledgments:** The author wishes to acknowledge the support and encouragement of Susana Asensio, Director of the Department of Urban Construction, who is also the former Director of Social Affairs for Guatemala City; Brady Greene, regional director for Vida Joven Guatemala; Rosario Burgos, Environmental Coordinator; Antonio Peña, Director of Nurseries; Lazaro Zamora, Director of the Central Wholesale Market (CENMA); and Kevin Gervais. In particular, this effort could not have been remotely possible without the hard work and commitment to effecting change in the lives of youths from Zone 3 in Guatemala City by eight outstanding graduate students from the University of Massachusetts; they are: Travis Shultz, Dan Shaw, Tamzeena Hutchinson, Adam Monroy, Brian Giggey, Seth Morrow, Megan Regan, and Jason Dell'Orfano.

**Photographic Credits:** The author took all of the photographs depicted in this research paper, unless otherwise noted.

**Conflicts of Interest:** The author declares no conflict of interest.

## Appendix A. Figures and Tables

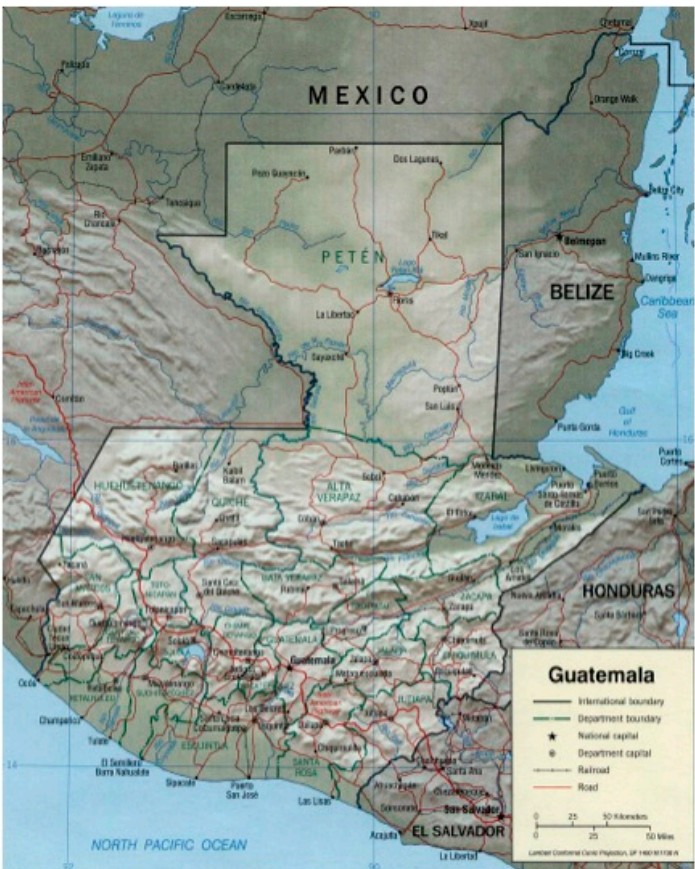

**Figure A1.** Guatemala is bordered by both the Pacific Ocean and the Caribbean Sea (Guia Geografico).

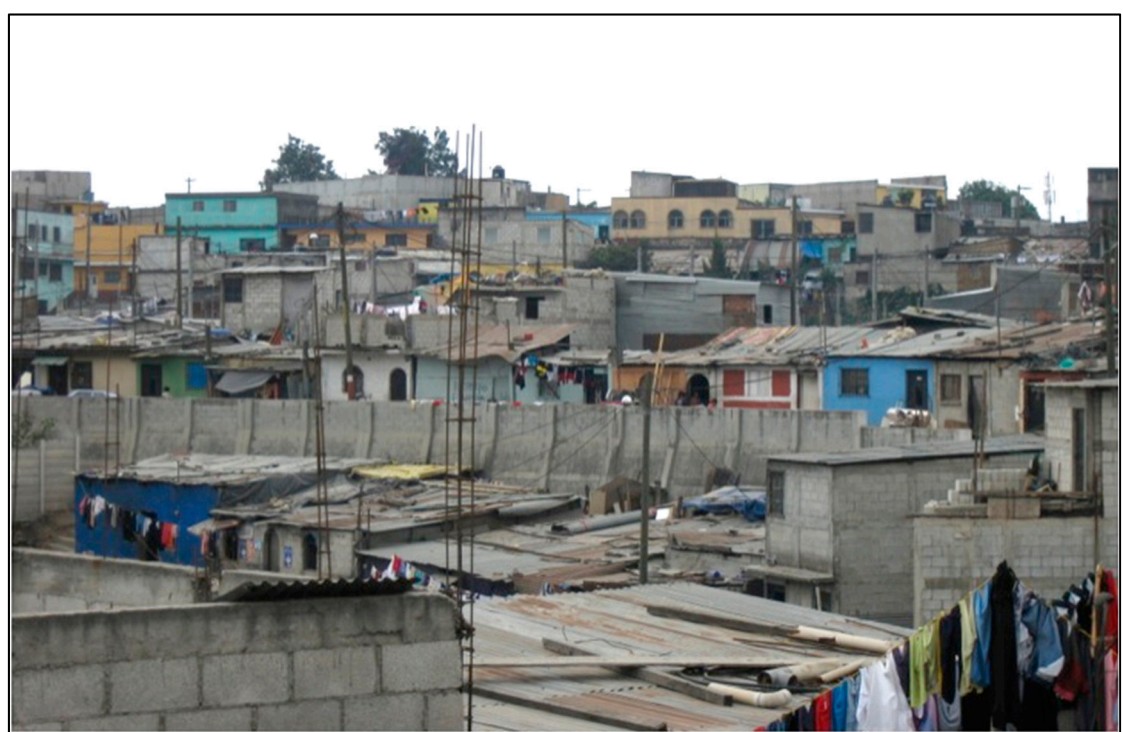

**Figure A2.** Squat dwellings constructed upon a former landfill that typify the Zone 3 neighborhood.

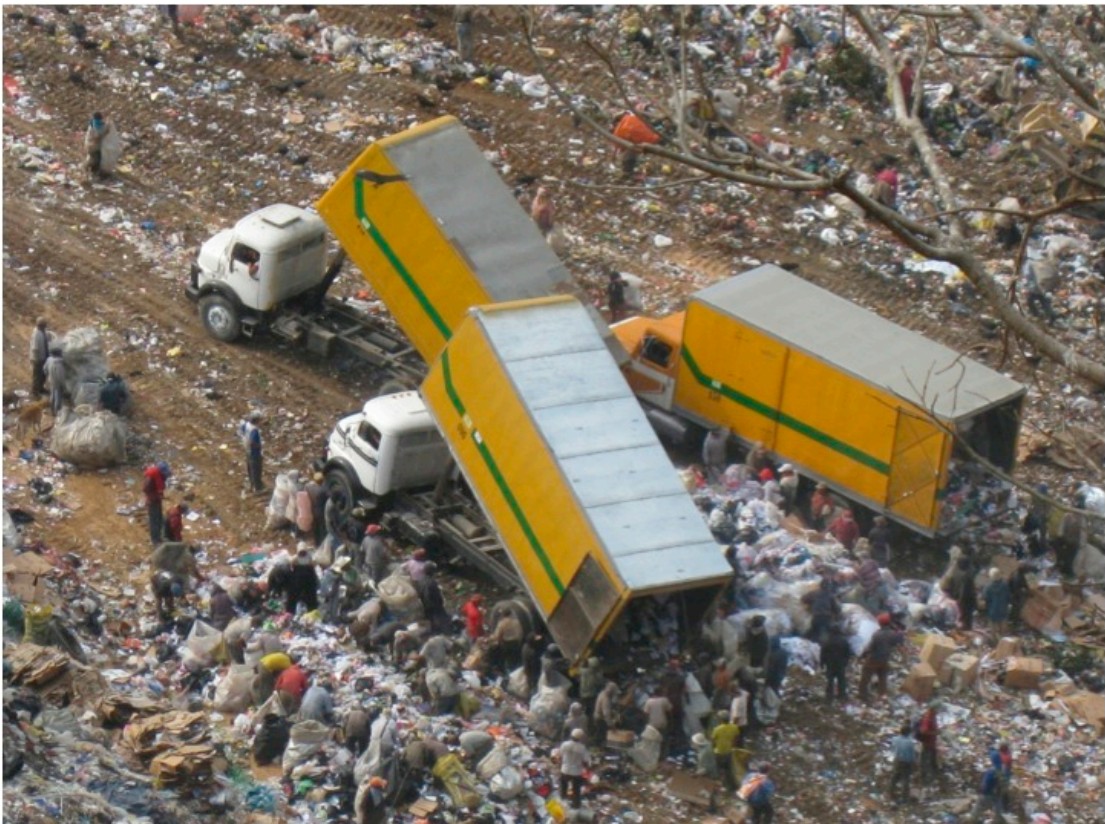

**Figure A3.** People scramble to gather what they can for recycling, food to eat, or other discarded waste items.

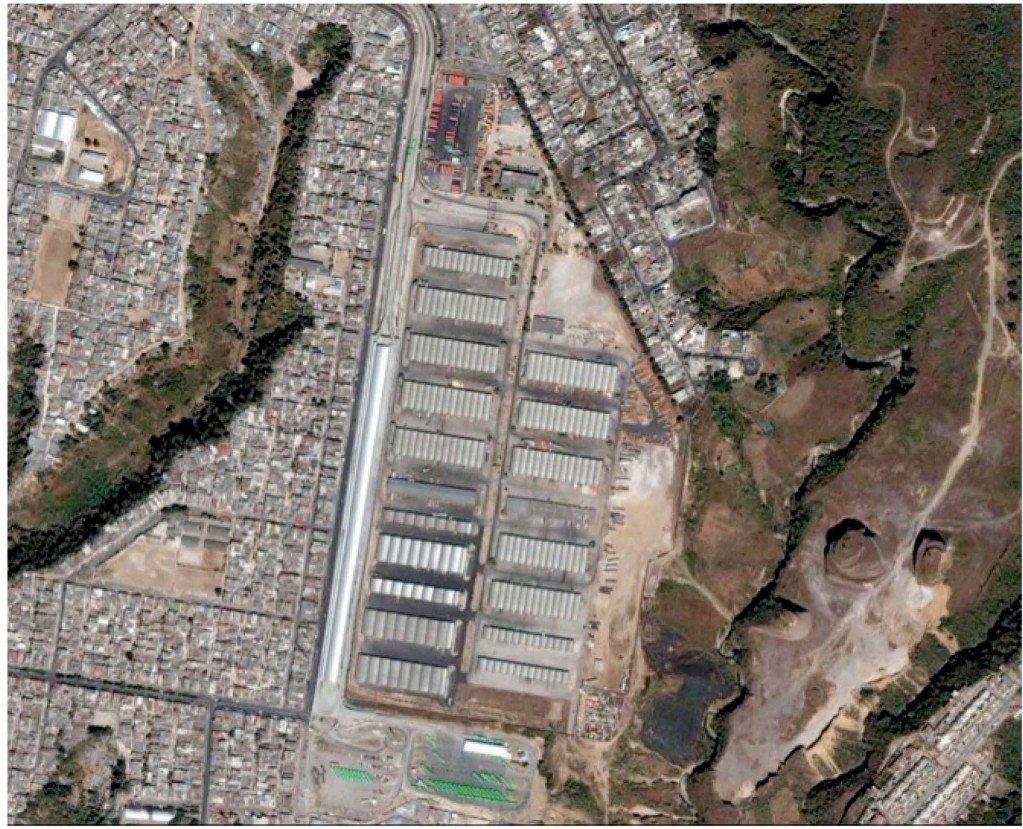

**Figure A4.** CENMA market (Central Market) is extensive. Note the covered open-air market structures (aerial image from Google Earth).

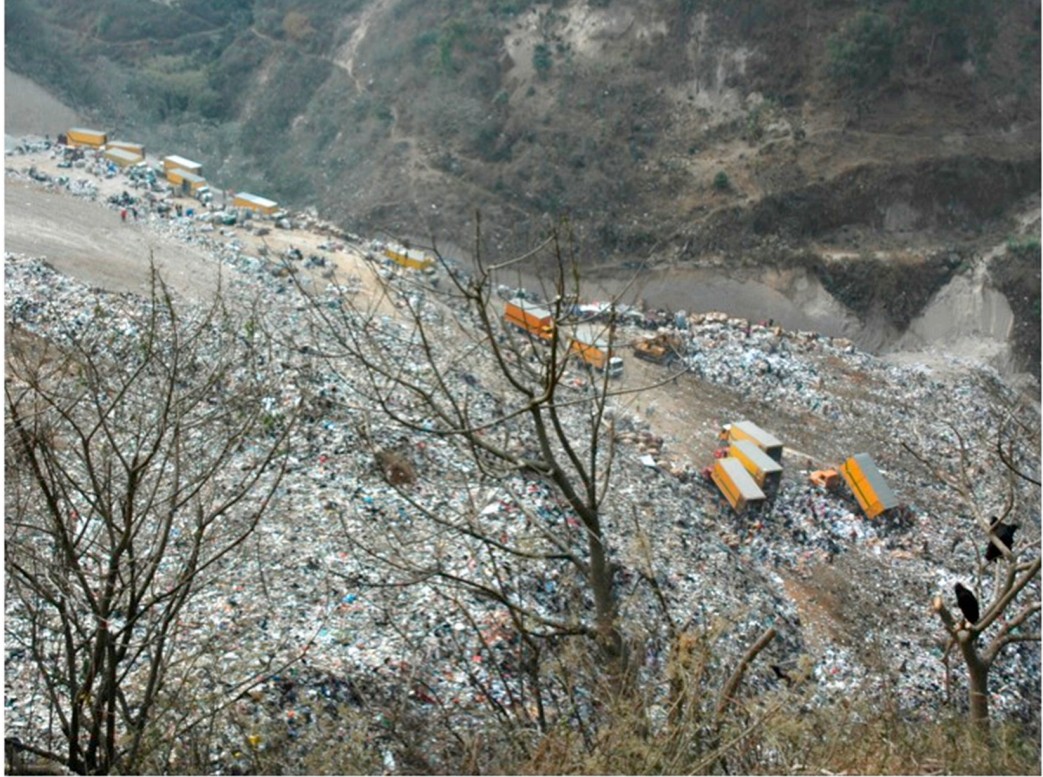

**Figure A5.** The landfill is expansive, very dangerous, prone to landslides, and emits methane gas from decomposing waste material.

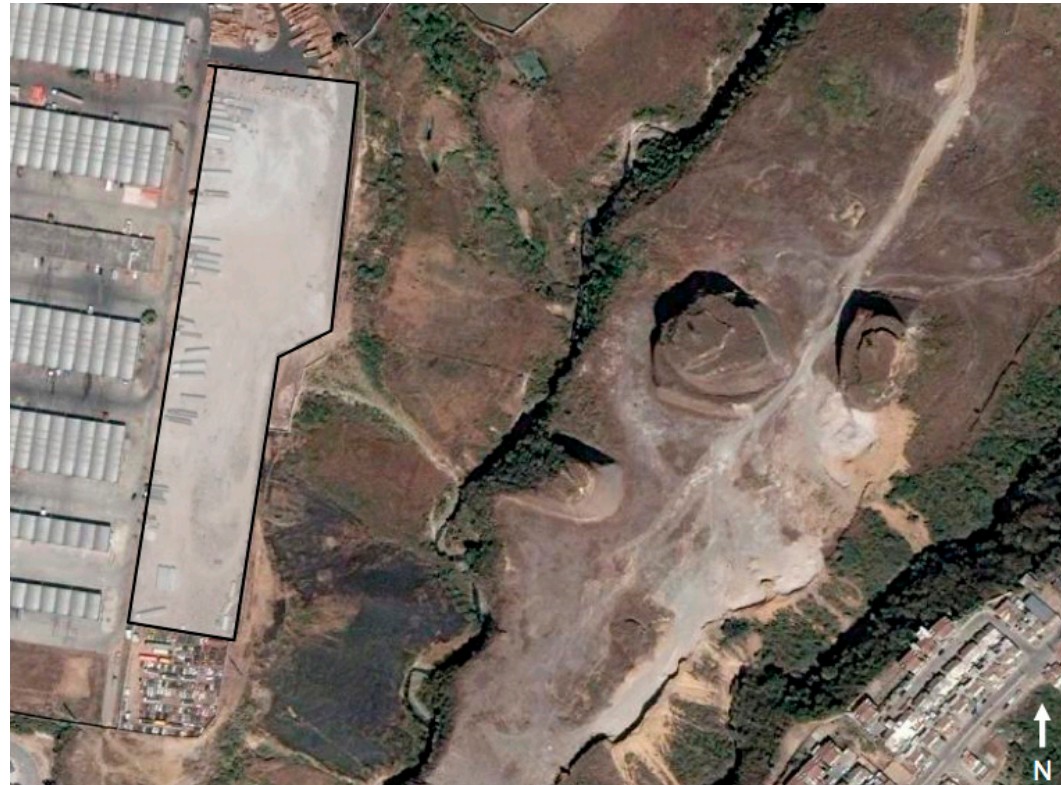

**Figure A6.** This aerial image of the large truck parking area located to the east of CENMA was promised to the University of Massachuesst team as the location of the compost operation (aerial image from Google Earth).

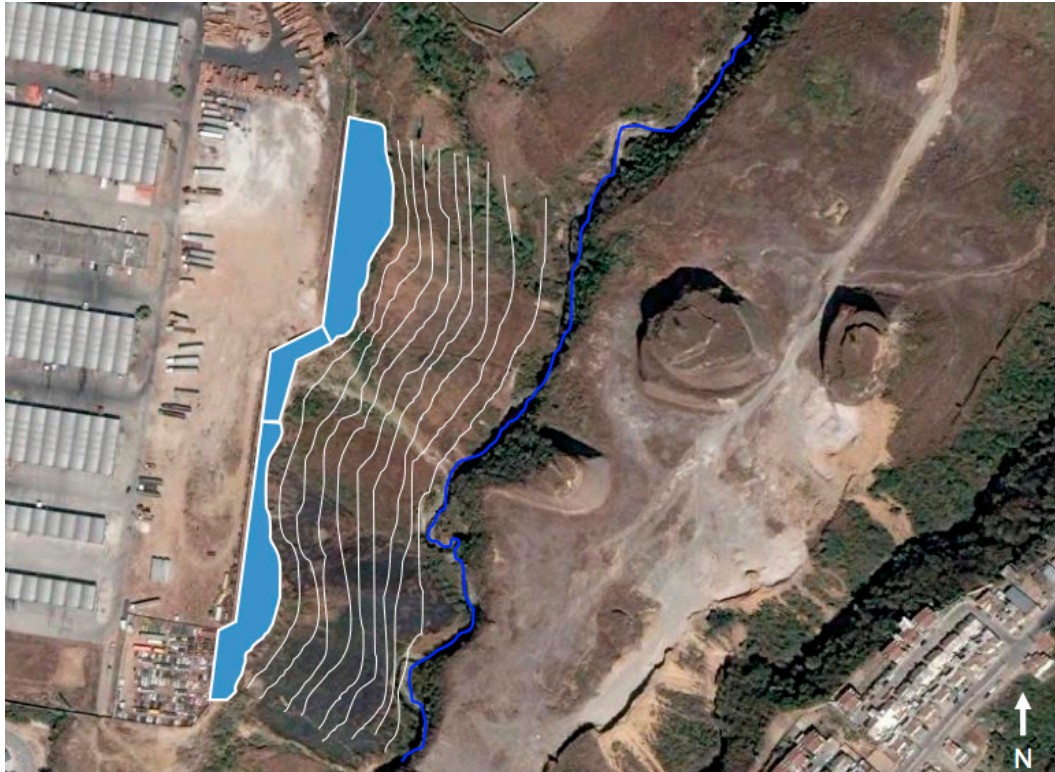

**Figure A7.** The blue shaded areas indicate a narrow strip of land that turned out to be the space made available for the compost plant design (aerial image from Google Earth).

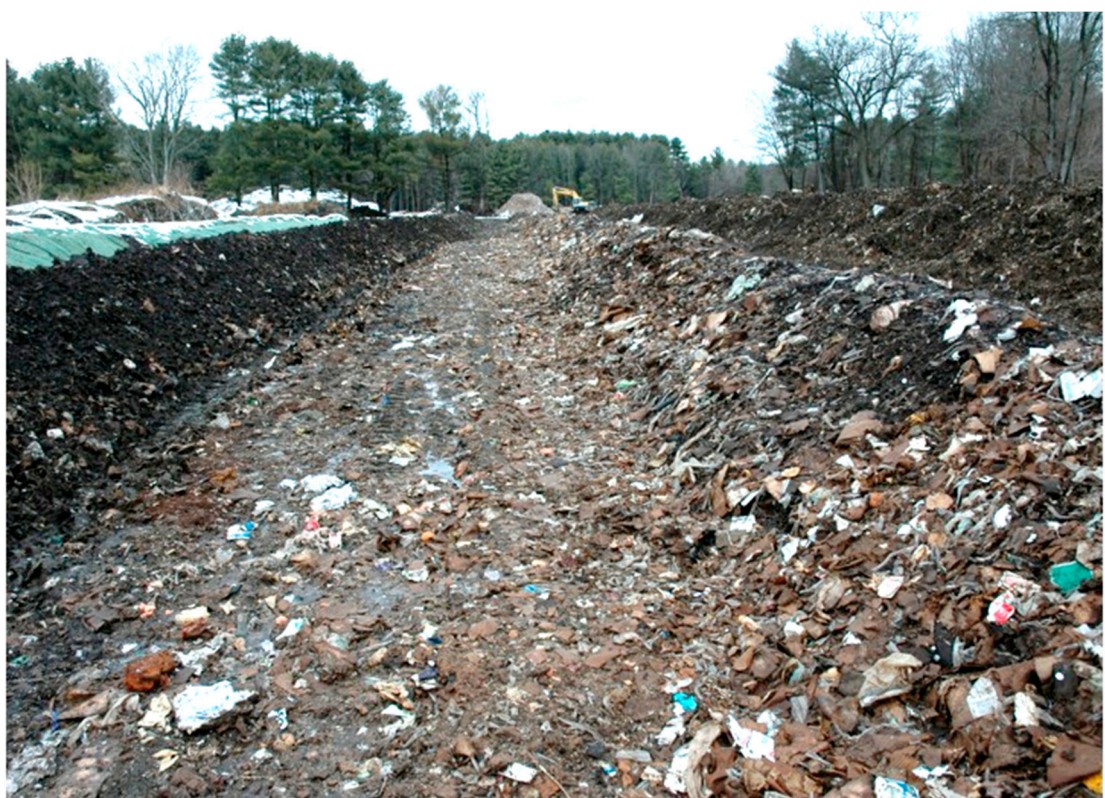

**Figure A8.** Windrows at a commercial compost operation in Greenfield, Massachusetts.

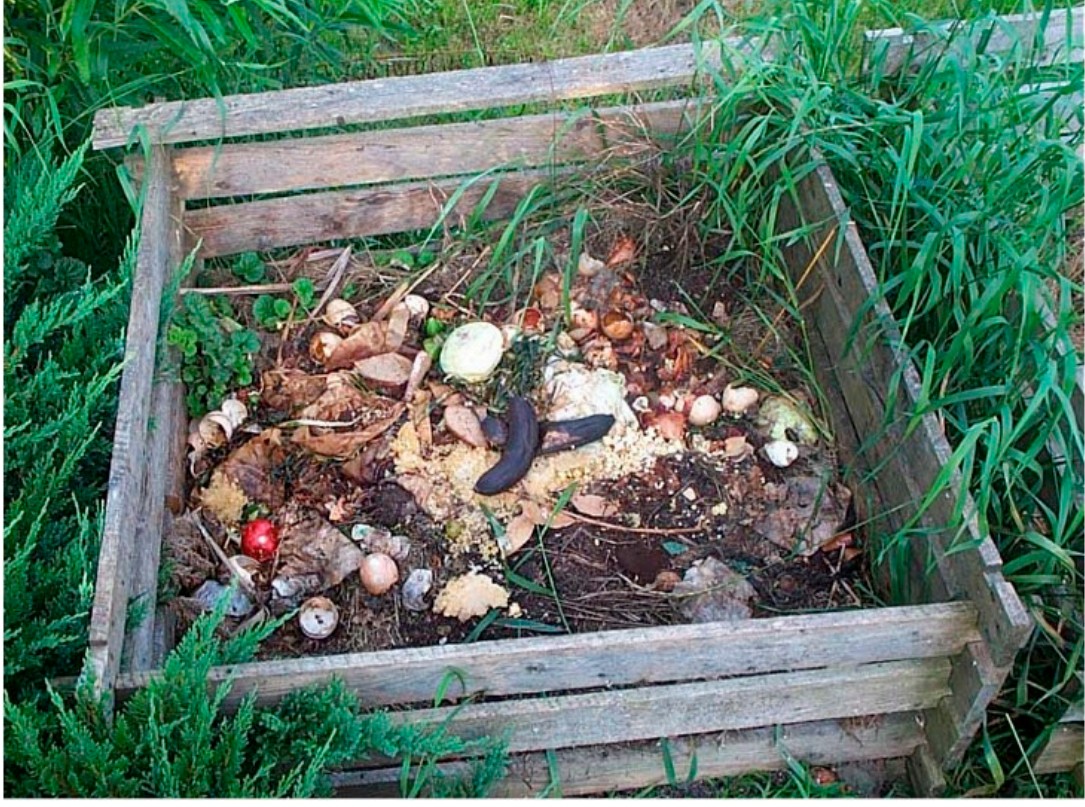

**Figure A9.** A back-yard compost bin typically employed by homeowners.

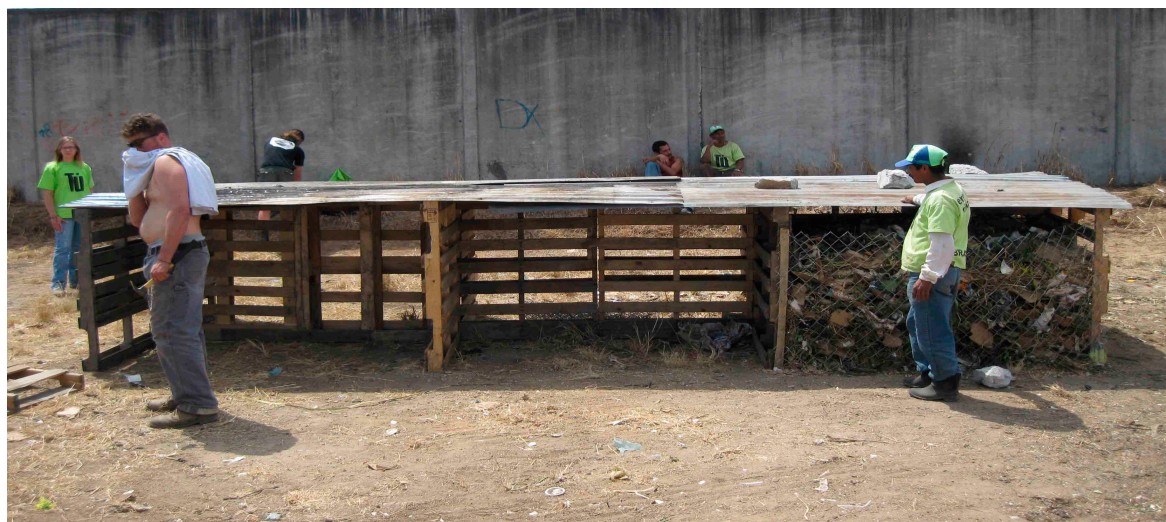

**Figure A10.** An image of the three-cell hybrid compost bin system adapted for the Guatemala test project.

**Table A1.** Data for each cell was carefully recorded to track decomposition, temperature, and odor.

| | | | | | | **Wendell** | | | |
|---|---|---|---|---|---|---|---|---|---|
| | | | | | | 1 = no smell | | 1 = dry | |
| | | | | | | 10 = smelly | | 10 = wet | |
| Date | Days | Height | Change | Temp | Change | Smell | Change | Humidity | Change |
| 22-Mar | | 30 | | 75 | | 3 | | 4 | |
| 27-Mar | 5 | 26 | −4 | 120 | 45 | 3 | 0 | 4 | 0 |
| 3-Apr | 7 | 23 | −3 | 130 | 10 | 2 | −1 | 3 | −1 |
| 13-Apr | 10 | 23 | 0 | 130 | 0 | 1 | −1 | 1 | −2 |

**Table A2.** Market segment analysis for compost produced.

| Segment | Description | Volume | Frequency | Comments (Including Where, Income, Reliability etc.) |
|---|---|---|---|---|
| Municipality | Government Projects | High | Seasonal/annual demand | Transportation implications low. Reliable demand and ability to pay. |
| Farmers | Rural Agriculture | High | Seasonal/annual demand | Possibility of unloading produce then loading compost. Reliable demand, low ability to pay. |
| Nurseries | Mostly urban flower or plant growers | Medium | Seasonal | Often a local market. Medium reliable demand and pay. |
| Households | Private gardeners growing vegetables or flowers | Low | Not strongly Seasonal, though peaks during spring | Local market, high ability to pay, packaging implications. |

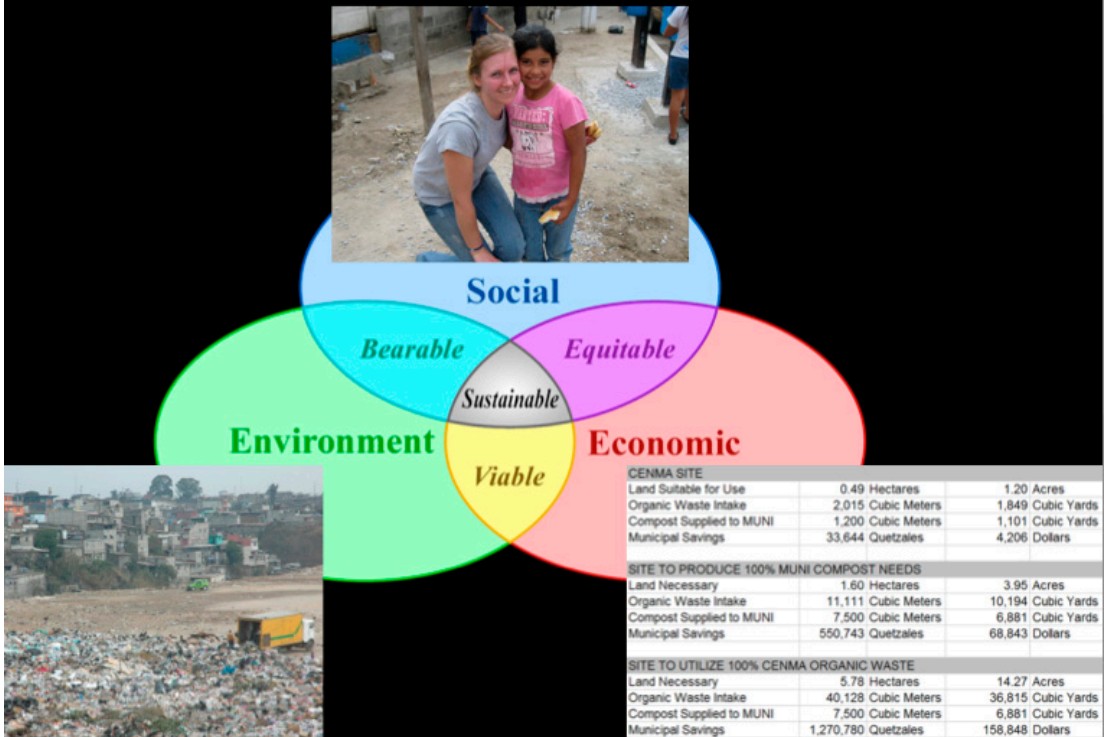

**Figure A11.** This applied research project in Guatemala City adhered to the three pillars of sustainability.

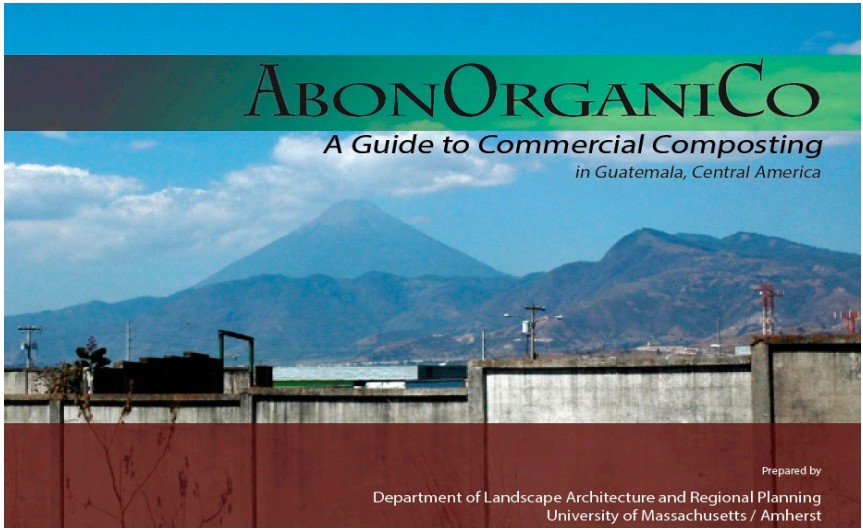

**Figure A12.** An operational guide prepared by the University of Massachusetts team detailed how to establish a non-profit company for employing at-risk youth from Zone 3 neighborhoods. Techniques for how to produce commercial-grade compost, along with site design guidelines, are also illustrated.

**Table A3.** Operational Manual for AbonOrganiCo.

| Table of Contents | | | | |
|---|---|---|---|---|
| **Introduction** | | | | |
| The Situation in Guatemala City | 6 | | | |
| Composting and Hope | 8 | | | |
| The Role of UMass Amherst | 9 | | | |
| **Part 1–Commercial Composting** | | | | |
| **Chapter I: Compost** | | **Chapter VI: Construction** | | |
| What It Is | 12 | Site Preparation | 42 | |
| How It Works | 13 | Vegetation Removal | 42 | |
| Managing the Compost Process | 14 | Grading the Site | 42 | |
| Compost or Fertilizer | 16 | Drainage Operations for Problem Areas | 44 | |
| **Chapter II: Composting Methods** | | Cell Construction | 44 | |
| Time Frame | 18 | Other Types of Cell Construction | 47 | |
| Mixing | 18 | **Chapter VII: Operation and Management** | | |
| In-cell or windrow Methods | 18 | Identify Brown Source | 50 | |
| Monitoring | 19 | Mixing Cup | 50 | |
| Turning the Piles | 19 | A : B : C Three Cell Method | 50 | |
| Oxygen | 20 | Combining Cells | 50 | |
| Water | 20 | Supervisor | 50 | |
| Screening | 21 | Labor | 51 | |
| Storage | 21 | Bobcat | 51 | |
| **Chapter III: Education** | | **Chapter VIII: Financial Business Plan** | | |
| Waste Separation | 24 | Executive Summary | 54 | |
| Compost and Usage | 24 | Legal Structure | 54 | |
| **Chapter IV: Marketing Guidelines** | | Products and Services | 55 | |
| Overview | 30 | Location | 56 | |
| Marketing Environment | 30 | Human Resources | 57 | |
| Competition and Pricing | 30 | Marketing Plan | 59 | |
| Product and Market | 31 | Start-Up Plan | 61 | |
| Promotional Strategy | 32 | 12-Month Profit and Loss Projection | 63 | |
| **Part 2–AbonOrganico** | | 4-Year Profit and Loss Projection | 66 | |
| **Chapter V: Site Design** | | The Search for Funding | 68 | |
| Current Site | 32 | Applying for Grants | 68 | |
| Current Site Plan | 36 | | | |
| Phrase II and III Site Possibilities | 38 | | | |
| | | Conclusion | 72 | |

**Table A4.** Promotional strategies.

| Word of Mouth | One Customer Tells Another about Your Product |
|---|---|
| Selling Technology | Face-to-face selling to the customer |
| Advertising | Communication through print, television, radio, billboard, etc. |
| Sales Promotion | Encourage people to buy more, "more for less", or trial periods. |
| Direct Marketing | Door-to-door, sales mail outs, telephone calls |
| Publicity | Press releases, public service events |
| Sponsorships | Where cash from one business supports an activity (e.g., sports) return for advertising and accociation with popular activities |
| Exhibitions | Displays promoting and demonstrating products |
| Identity | Developing a logo or catch phrase establishes professionalism and differentiates form competitors |
| Packaging | Important in attracting first time customers, and also getting marketing information across |

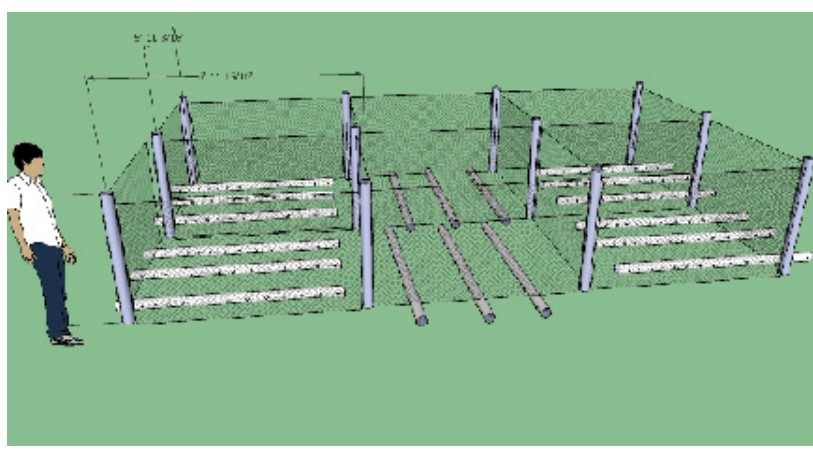

**Figure A13.** Schematic illustrating the cell construction using metal stakes and wire fencing from the Operational Manual produced by students. Note the perforated piping for movement of air that will accelerate decomposition (illustration by Seth Morrow).

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
