# Peer review of "Reflections on Service Learning for a Circular Economy Project in a Guatemalan Neighborhood, Central America"

_sustainability, doi:10.3390/su11174776_

Round 1

Reviewer 1 Report

The present paper is built upon two research questions, namely i) could usable compost be made from the organic? and; ii) could this effort create jobs and train workers?. It is a relevant topic which worth to be investigated. It is a quite interesting manuscript. However, several challenges are required:

The main issue is the lack of scientific reference to literature. A strong effort is required to make the paper scietific.

Abstract has inappropriate structure. I suggest to answer the following aspects: - general context - novelty of the work - methodology used - main results 

Introduction presents interesting information. However, does not succeed to frame the framework within relevant literature. Scientific literature is almost absent. This section need to be reinforced: You could for example emphasize the role of circular economy also looking at social aspects. I would suggest to use this section to discuss about the relevance of waste materials for the application of circular economy principle. The circular economy approach has the goal to make better use of resources/materials through reuse, recycling and recovery, and also to minimise the energy and environmental impact of resource extraction and processing (see Falcone and Imbert, 2018 (“Social life cycle approach as a tool for promoting the market uptake of bio-based products from a consumer perspective”) Basically, it denotes new business models which aims at creating industrial systems that are purposely restorative, by reducing unintended negative consequences on the environment of production and consumption processes (Millar et al 2019- “The Circular Economy: Swings and Roundabouts?”;  Morone et. Al 2019 “How to promote a new and sustainable food consumption model: A fuzzy cognitive map study”, Goyal 2019 “Reducing Waste in Circular Economy” ).

The research methodology seems underdeveloped. What are exactly the variables? Methods should be described in detail. Indeed, I think the research procedure could be much more clearly described by means of a diagram also highlighting its potential and limit.

Results need to be discussed in light of literature.

 Conclusions are extremely succinct. I suggest to authors to propose policy directions. Link with future lines of research should look at financial sustainability of projects.  A clear example is “green fiance”. Some relevant contribution to start with are:

Some suggestions are: 

https://www.sciencedirect.com/science/article/pii/S0956053X18304823

https://www.mdpi.com/2071-1050/11/2/517

https://www.sciencedirect.com/science/article/pii/S0921800917317573 

https://www.sciencedirect.com/science/article/pii/S0040162517306716 

https://www.sciencedirect.com/science/article/pii/S0040162517306820

Author Response

The main issue is the lack of scientific reference to literature. A strong effort is required to make the paper scietific.

Abstract has inappropriate structure. I suggest to answer the following aspects: - general context - novelty of the work - methodology used - main results 

Thank you for your appropriate and well crafted review comments on my paper. I have taken your comments and suggestions and done a significant re-crafting of the paper and its structure to meet your recommendations for improved scientific scholarship. Whereas this paper was not initially intended to present the results of a purely scientific investigation but rather an applied action research based study and project, the “take aways” that were learned by students and the PI (myself) were unexpected, yet profound. As such, my feelings were that it would be valuable to present this to the readers of the journal. The responses to this work when I have presented it at a variety of scientific conferences showed that the audience was tremendously enthusiastic about the results of the study and felt that more applied hands on research is needed for work focusing on the key elements of sustainability in action. As such I have included references to the literature used during the study that explored how to best apply Action Research and Community Service Learning within the framework of setting up a company focused upon social sustainability first.

Introduction presents interesting information. However, does not succeed to frame the framework within relevant literature. Scientific literature is almost absent. This section need to be reinforced: You could for example emphasize the role of circular economy also looking at social aspects. I would suggest to use this section to discuss about the relevance of waste materials for the application of circular economy principle. The circular economy approach has the goal to make better use of resources/materials through reuse, recycling and recovery, and also to minimise the energy and environmental impact of resource extraction and processing (see Falcone and Imbert, 2018 (“Social life cycle approach as a tool for promoting the market uptake of bio-based products from a consumer perspective”) Basically, it denotes new business models which aims at creating industrial systems that are purposely restorative, by reducing unintended negative consequences on the environment of production and consumption processes (Millar et al 2019- “The Circular Economy: Swings and Roundabouts?”;  Morone et. Al 2019 “How to promote a new and sustainable food consumption model: A fuzzy cognitive map study”, Goyal 2019 “Reducing Waste in Circular Economy” ).

The Introduction has been re written whereby principles of circular economy are discussed (Geissdoerfer et al 2017; Petit-Box et al 2918; Kalmykova el al 2018; and Korhonen 2018), etc. During the work in Guatemala, we were thinking that the concept for making compost from green material that arrived in the city from rural farms would allow us to then load completed compost onto the now empty trucks for their return trip to the rural landscape, suggesting a closed loop, or circular system. In practice, we did not achieve this as the farmers were not willing to pay for the compost, only the municipal government in Guatemala City and some local landscaping firms. Yes the social aspects, in particular Social Sustainability, as one of the key “pillars” was the driving engine of this overall work and research effort. I have reinforced this in both the Introduction and Section 2: Materials and Methods. All of your suggested literature was very helpful to read and it has stimulated my thinking for a much more narrowly targeted future paper with colleagues from my Faculty of Business.  

The research methodology seems underdeveloped. What are exactly the variables? Methods should be described in detail. Indeed, I think the research procedure could be much more clearly described by means of a diagram also highlighting its potential and limit.

The description of the research methodology has been expanded and re-crafted to include the work employing Action based Research as applied to the project. Again, this endeavor was not initiated as a purely scientific research project to test different methodologies within a given context and determine their effectiveness in solving X, Y or Z. Nor did it involving using one given methodology to be tested in a variety of scenarios to see how the results might be similar or different and why might this might be the case. It involved trying to make some real and meaningful change to a minority population, embracing social sustainability in combination with action research through the lens of community service learning. I recognize that sometimes when you try to incorporate a broader range of techniques on a multi-varied setting, being clear on what is learned and do the results, in fact, really reflect on the methods employed can be more grey or vague in their outcome(s). However the methods used are sound and supported in the literature, albeit perhaps less scientific and leaning more toward design and social sciences. I have greatly expanded the 3 Results and 4 Discussion to support this.

Results need to be discussed in light of literature.

Both Results and Discussion now achieve this.

Conclusions are extremely succinct. I suggest to authors to propose policy directions. Link with future lines of research should look at financial sustainability of projects.  A clear example is “green fiance”. Some relevant contribution to start with are:

Some suggestions are: 

https://www.sciencedirect.com/science/article/pii/S0956053X18304823

https://www.mdpi.com/2071-1050/11/2/517

https://www.sciencedirect.com/science/article/pii/S0921800917317573 

https://www.sciencedirect.com/science/article/pii/S0040162517306716 

https://www.sciencedirect.com/science/article/pii/S0040162517306820

Reviewer 2 Report

Overall I've read an interesting but improvable manuscript.

The topic dealt within the paper is very timely, yet, the style of writing is not a scientific one. As a reader, you have the sensation of reading a blog rather then an scientific paper. This diminishes the amount of effort made by the author to carry out the research. In my opinion, the entire article should be rewritten with a focus on the Guatemala example of using wastes for compost business, not on the people encountered during the demarche. Also, I suggest the author to remove some of the figures.

Furthermore, the author should pay much more attention to journal's template.

Author Response

The topic dealt within the paper is very timely, yet, the style of writing is not a scientific one. As a reader, you have the sensation of reading a blog rather then an scientific paper. This diminishes the amount of effort made by the author to carry out the research. In my opinion, the entire article should be rewritten with a focus on the Guatemala example of using wastes for compost business, not on the people encountered during the demarche. Also, I suggest the author to remove some of the figures.

Furthermore, the author should pay much more attention to journal's template.

Thank you for your appropriate and well crafted review comments on my paper. I have taken your comments and suggestions and done a significant re-crafting of the paper and its structure to meet your recommendations for improved scientific scholarship. Whereas this paper was not initially intended to present the results of a purely scientific investigation but rather an applied action research based study and project, the “take aways” that were learned by students and the PI (myself) were unexpected, yet profound.

I have re-crafted all of the sections to more closely follow the journal template. In particular, Section 2, Materials and Methods now include a greater amount and divsersity of scientific and scholarly references. Section 3, Results and Section 4, Discussion are much more complete and reflect upon the intentions expressed in the Introduction.

Whereas I recognize that to mention the names of important people who lead toward the overall research effort is not typical of a scientific paper; without them the research effort to make compost and involve marginalized youth in Guatemala City would not have happened. If you feel that their names should be removed and just their professional titles used, I can do that if the journal editor believes it appropriate.

Because this effort has been built upon the application of Action Research, a science that is not purely quantitative or qualitative in practice, the scholarship may seem as different shades of grey and not specifically as quantitative results, the results were far from this. Because the educational pedagogy emphasized Community Service Learning, the results were much more than ‘feel good’ outcomes but resulting in real and meaningful transformation on the part of the students and those lives in Guatemala City who we made an impact upon. Yes, some of the images/figures can be removed, however I feel that they help to visually portray the current situation and context.

Round 2

Reviewer 1 Report

Dear Authors,
thanks for the new version of the manuscript. Although you tried to address some first round suggestions, you did not account properly at my revisions comments.
The introduction fails to set out any specific interest to a broader audience. There is nothing more than a sort of putting forward the topic and the claim of a research question; paper structure is missing.
I strongly believe that a literature section is needed. What about contribution to relevant literature? Where is the research gap? In this case, it is not sufficiently supported. I suggest the authors to follow a specific narrative where circular economy ( https://www.sciencedirect.com/science/article/pii/S0959652617320425) and social aspects needs to be stressed (https://onlinelibrary.wiley.com/doi/full/10.1002/csr.1791)
You did not answer to my concerns about methods. Please read carefully my previous comments. I still suggest to use a diagram to better explain methods highlighting its potential limit. 
Finally, Conclusions are still extremely succinct. I suggest to authors to propose practical implications. Link with future lines of research should look at financial sustainability of projects. The role of green and social finance is gathering increasing attention. References:
https://www.sciencedirect.com/science/article/pii/S0959652617321650
https://www.mdpi.com/2071-1050/11/2/517
https://www.sciencedirect.com/science/article/pii/S0275531917306414

Author Response

The introduction has been rewritten to appeal to a broader audience as you have suggested. It now includes a much more detailed literature review, particularly addressing circular economy, its theories, contested themes, and then makes a connection to how CE within the context of making compost from market waste supported principles of a closed – loop system.

Methods in this applied research project were not as scientific as is typically conducted in a more traditional investigation where systematic variables may exist to support of cause the failure of an experiment or tested hypothesis. I have tried to develop a diagram illustrating this but believe that the now clarified Introduction, Results and Conclusion will help to present the approach more clearly.

The results do now reflect back upon the literature and its pedagogical framework of CSL and Action base Research.

Conclusions have been expanded to address the success and limitations of the results from this research. Green financing has been address in the literature review included in the introduction.

Reviewer 2 Report

Despite some improvements in the description and some good effort in trying to improve the state of art, the author just partially answered to my feedback. The scientific writing style is still missing. Most figures are not significant for the content. Mentioning the names of the experts or the people the author interviewed is not the problem, yet, the manner of describing it within the context of the article. I think that the title itself is not supported by the article. 

Author Response

Scientific writing style has been presented in the new Introduction and associated literature review.

I have eliminated approximately 10 Figures and kept only those to help illustrate key points described in the text.

Names of individuals interviewed and or included in the research have been removed and the author only refers to himself as such.

The title of the paper as been changed and new abstract developed.

Round 3

Reviewer 1 Report

Dear author,

the paper is much improved. Congratulations

Reviewer 2 Report

Changes have been made to the manuscript. I consider it now suitable for publication within Sustainability